# Discovery of 4-Anilinoquinolinylchalcone Derivatives as Potential NRF2 Activators

**DOI:** 10.3390/molecules25143133

**Published:** 2020-07-08

**Authors:** Yu-Tse Kao, Yi-Siao Chen, Kai-Wei Tang, Jin-Ching Lee, Chih-Hua Tseng, Cherng-Chyi Tzeng, Chia-Hung Yen, Yeh-Long Chen

**Affiliations:** 1Department of Medicinal and Applied Chemistry, College of Life Science, Kaohsiung Medical University, Kaohsiung 807, Taiwan; u104850010@Kmu.edu.tw (Y.-T.K.); tzengch@kmu.edu.tw (C.-C.T.); 2Program in Environmental and Occupational Medicine, College of Medicine, Kaohsiung Medical University, Kaohsiung 807, Taiwan; wl01420323@gmail.com; 3Graduate Institute of Natural Products, College of Pharmacy, Kaohsiung Medical University, Kaohsiung 807, Taiwan; jclee@kmu.edu.tw; 4School of Pharmacy, College of Pharmacy, Kaohsiung Medical University, Kaohsiung 807, Taiwan; dadaking1107@gmail.com (K.-W.T.); chihhua@kmu.edu.tw (C.-H.T.); 5Department of Biotechnology, College of Life Science, Kaohsiung Medical University, Kaohsiung 807, Taiwan; 6Drug Development and Value Creation Research Center, Kaohsiung Medical University, Kaohsiung 807, Taiwan; 7Department of Medical Research, Kaohsiung Medical University Hospital, Kaohsiung Medical University, Kaohsiung 807, Taiwan

**Keywords:** 4-anilinoquinolinylchalcone derivatives, nuclear factor erythroid-2-related factor 2 (NRF2) activators, cancer chemopreventive agent

## Abstract

Activation of nuclear factor erythroid-2-related factor 2 (NRF2) has been proven to be an effective means to prevent the development of cancer, and natural curcumin stands out as a potent NRF2 activator and cancer chemopreventive agent. In this study, we have synthesized a series of 4-anilinoquinolinylchalcone derivatives, and used a NRF2 promoter-driven firefly luciferase reporter stable cell line, the HaCaT/ARE cells, to screen a panel of these compounds. Among them, (*E*)-3-{4-[(4-acetylphenyl)amino]quinolin-2-yl}-1-(4-fluorophenyl)prop-2-en-1-one (**13b**) significantly increased NRF2 activity in the HaCaT cell with a half maximal effective concentration (EC_50_) value of 1.95 μM. Treatment of compound **13b** upregulated HaCaT cell NRF2 expression at the protein level. Moreover, the mRNA level of NRF2 target genes, heme oxygenase-1 (HO-1), glutamate-cysteine ligase catalytic subunit (GCLC), and glucose-6-phosphate dehydrogenase (G6PD) were significantly increased in HaCaT cells upon the compound **13b** treatment. The molecular docking results exhibited that the small molecule **13b** is well accommodated by the bound region of Kelch-like ECH-associated protein 1 (Keap1)-Kelch and NRF2 through stable hydrogen bonds and hydrophobic interaction, which contributed to the enhancement of affinity and stability between the ligand and receptor. Compound **13b** has been identified as the lead compound for further structural optimization.

## 1. Introduction

The nuclear factor erythroid-2-related factor (NRF2) is a transcription factor that is sensitive to oxidative stress. In normal conditions, NRF2 activity is suppressed by Kelch-like ECH-associated protein 1 (Keap1), which is a substrate adaptor for Cullin-3 (CUL3) [1,2,3]. NRF2 will then be ubiquitinated by CLU3 and be degraded by proteasome. Under oxidative stress conditions, reactive oxygen species (ROS) or electrophiles oxidize key sensor cysteine residues in Keap1 and change its conformation, which disrupts the interaction between Keap1 and NRF2 [4]. NRF2 then can enter the nucleus and forms dimer with sMaf. This dimer binds to the antioxidant response element (ARE) and activated a variety of antioxidant proteins and detoxification enzymes, which play important roles in antioxidant response and cancer prevention [5,6]. Therefore, activation of NRF2 has long been considered as a potential strategy to prevent oxidative damage caused by UV irradiation or carcinogens exposure [7].

In addition to Keap1, the activation of different protein kinases has been shown to activate NRF2 [8]. The NRF2-regulated genes include almost all of the relevant antioxidants and cytoprotective genes such as heme oxygenase-1 (HO-1), NAD(P)H/quinone oxidoreductase 1 (NQO1), glutamate cysteine ligase modifier subunit (GCLM), γ-glutamyl cysteine synthase, glutathione peroxidase (GPx), and several members of the glutathione S-transferase family that express an ARE in their promoter [9]. Small molecules that activate NRF2 signaling are being investigated as potential anticancer or anti-inflammatory agents [10]. A wide variety of dietary and synthetic compounds that function as potent inducers of ARE-regulated gene expression have been shown to exert chemopreventive activities, e.g., sulforaphane [11], dithiolethione [12], curcumin [13], and caffeic acid phenethyl ester (CAPE) [14]. It is notable that both curcumin and CAPE, which bear a α,β-unsaturated ketone moiety, can function as Michael acceptors, and therefore modify cysteine thiols present in Keap1. Chalcones are Michael acceptors and constitute an important group of natural products belonging to the flavonoid family [15].

Certain chalcone derivatives have been reported as potent NRF2 activators (Figure 1). For examples, (*E*)-1-(2-methoxyphenyl)-3-[2-(trifluoromethyl)phenyl]prop-2-en-1-one (Compound **1**) [16] was found to activate the expression of GCLM and NQO1 in the small intestine with 3-fold and 5-fold higher respectively compared to sulforaphane. Wu et al. have revealed that (*E*)-2-methoxy-4-(3-(4-methoxyphenyl)-3-oxoprop-1-en-1-yl)phenyl acrylate (compound **2**) [17] increased the expression of NRF2-dependent antioxidant genes, such as glutamate-cysteine ligase complex (GCLC) and HO-1, and their corresponding proteins, together with counteracting neuronal death induced by H_2_O_2_ in PC-12 cells. Lee et al. demonstrated that (*E*)-1-(2,5-dihydroxy-4-methoxyphenyl)-3-(4-hydroxyphenyl)prop-2-en-1-one (compound **3**) [18] could increase the antioxidant endogenous defense via the Keap1/NRF2 pathway. Lee et al. found that (*E*)-1-phenyl-3-(2,4,6-tris(methoxymethoxy)phenyl)prop-2-en-1-one (compound **4**) [19] prevented TNF-α-induced inflammation in HT-29 cells through the nuclear translocation of NRF2 and a consequent increase of HO-1 expression. Kachadourian et al. revealed that (*E*)-1-(2,5-dihydroxyphenyl)-3-phenylprop-2-en-1-one (compound **5**) [20] was capable of increasing the NRF2–ARE activity and glutathione (GSH) levels. Pinner et al. discovered that flavokawains A [21], a methoxylated chalcone from Kava (*Piper methysticum*), activated NRF2, increasing the expression of antioxidant and heat shock (Hsp) response genes, in HepG2 cells.

The quinoline ring can be recognized in various biologically active compounds with clinical applications. The most noteworthy example is neratinib, an irreversible tyrosine kinase inhibitor, which is used as an adjuvant therapy in people with early stage breast cancer in which HER2 is overexpressed. Ciprofloxacin is another example of quinoline derivative, which inhibits DNA gyrase activity and clinically used for the treatment of bacteria infection. Other examples of quinoline derivatives such as quinine and primaquine were found to possess antimalaria activity. Therefore, preparation and extensive biological evaluations on quinoline derivatives have continuously attracted our attention.

Recently, we have synthesized certain 2-substituted 3-arylquinoline derivatives and the evaluation of their anti-inflammatory effects in LPS-activated murine macrophage cell line J774A.1 cells. 2-(4-Fluoro-benzoyl)-3-(3,4,5-trimethoxyphenyl)quinoline significantly decreases the secretion of pro-inflammatory cytokines (TNF-α and IL-6) in the LPS-activated macrophages [22]. The present report describes the synthesis of novel hybrid molecules, which bears both quinoline and chalcone moieties and evaluate for their NRF2 activating activities.

## 2. Chemistry

2-Methylquinolin-4-ol (**6**) was treated with POCl_3_ to give the intermediate 4-chloro-2-methylquinolin (**7**), which was then oxidized with selenium oxide to afford 4-chloroquinoline-2-carbaldehyde (**8**) [23,24], as described in Scheme 1. Aldol condensation of compound **8** with appropriately substituted acetophenones under a basic condition at 0 °C gave quinolinylchalcone derivatives, compounds **9–11**, in a yield of 50–60%. These quinolinylchalcones were obtained as geometrically pure and with *trans*-configuration (*J*
_Ha–Hb_ = 15.2–16.0 Hz) [25] of the sole (*E*)-form isomers. The desired 4-anilinoquinolinylchalcone derivatives **12**–**14** were synthesized by the reaction of compounds **9–11** respectively with various 4-substituted anilines under acidic condition. The structures of **12–14** were determined by NMR spectrometer (spectra data can be found in Appendix A) and compound purity is determined by high performance liquid chromatography (HPLC), and all final test compounds were >95% purity.

## 3. Biological Results and Discussion

### 3.1. Novel 4-Anilinoquinolinylchalcone Derivatives Induced ARE-Driven Luciferase Activity in HaCaT/ARE Cells

In this study, we used a stable ARE-driven reporter system to screen the potential NRF2 activators [26]. Luciferase activities were determined after cells treated with test compounds at a concentration of 10 μM for 24 h. *tert*-Butyl hydroquinone (*t*-BHQ), a well-known NRF2 activator in ARE-controlled gene transcription, showed 1880% induction of ARE-driven luciferase activity, was used as the positive control. In addition, flavokawain A, which was identified as a potential chemopreventative or chemotherapeutic agent by Pinner et al. [21], was adapted as a second positive control, for the 1-phenylprop-2-en-1-one derivatives (R_1_ = H), (*E*)-1-phenyl-3-[4-(phenylamino)quinolin-2-yl]prop-2-en-1-one (**12a**, R_2_ = H, 7% of relative NRF2 activity) showed no induction of the luciferase activity while its 4-acetyl derivative (**12b**, R_2_ = COMe, 394% of relative NRF2 activity) was weakly active as shown in Table 1. Oximination of **12b** decreased NRF2 inductive activity in which the potency decreased in an order **12b** > **12d** (287%) > **12e** (95%) > **12f** (26%). The 4-carboxylic acid derivative (**12c**, R_2_ = COOH, 75%) was less active than its 4-acetyl counterpart (**12b**, R_2_ = COMe, 394%) but more active than the parent (**12a**, R_2_ = H, 7%) while amide (**12g**, 9%) and ester (**12h**, 32%) derivatives were inactive. The same structure–activity relationships were observed for the 1-(4-fluorophenyl)prop-2-en-1- one derivatives (R_1_ = F) in which the NRF2 inductive activity decreased in an order **13b** (R_2_ = COMe, 2205%) > **13c** (R_2_ = COOH, 147%) > **13a** (R_2_ = H, 6%). Compound **13b** was more active than both positive controls, *t*-BHQ (1880%) and flavokawain A (2082%). Oximination of **13b** gave **13d**–**13f,** which exhibited less NRF2 inductive effects, compounds **13d** (1697%) and **13f** (1850%) were comparable to the positive *t*-BHQ while compound **13e** (2145%) was more active. The ester derivative **13h** (1767%) was approximately equal active to *t*-BHQ in the NRF induction while the amide counterpart **13g** (5%) was inactive. For the 1-(4-methoxyphenyl)prop-2-en-1-one derivatives (R_1_ = OMe), the same structure–activity relationships have also been obtained in which the NRF2 inductive activity decreased in an order **14b** (937%) > **14c** (212%) > **14a** (7%). Oximination of **14b** decreased NRF2 inductive activity in which the potency decreased in an order **14b** > **14d** (229%) > **14e** (193%) > **14f** (6%). The ester derivative **14h** (127%) was weakly active in the NRF induction while the amide counterpart **14g** (5%) was inactive. Our results indicated the best substituent for R_2_ is the acetyl group in all the 4-anilinoquinolinylchalcone derivatives tested and (*E*)-3-{4-[(4-acetylphenyl)amino]quinolin-2-yl}-1-(4-fluorophenyl)prop-2-en- 1-one (**13b**, R_1_ = F, R_2_ = COMe) was the most potent NRF2 activator, which enhanced ARE promoter activity to 2205%.

The cell viability of 4-anilinoquinolinylchalcone derivatives is also summarized in Table 1. Most of them exhibited moderate cytotoxicity with the survival rate of less than 90% at 10 µM concentration. However, compound **13b** was relatively non-cytotoxic, with the survival rate of 96% at the same concentration. Therefore, compound **13b** and its analogs, **12b** and **14b,** were selected for further evaluations. Results from Figure 2 indicated compounds **12b**, **13b**, and **14b** induced NRF2 promoter activity in a dose-dependent manner with EC_50_/Emax at 1.95 ± 0.24 μM/1095% ± 48%, 0.85 ± 0.04 μM/364% ± 15%, and 1.58 ± 0.15 μM/393% ± 24%, respectively. We have also compared the cytotoxicity of these three compounds. A slight cytotoxicity of compounds **13b** (CC_50_ > 100 μM) was observed, while compound **14b** and **12b** are more toxic to keratinocyte, and the CC_50_ of these two compounds are 28.4 μM and 9.3 μM, respectively as shown in Figure 3.

### 3.2. The Effect of Compound 13b on the NRF2 Pathway

To further evaluate the effect of compound **13b** on NRF2 activation, the mRNA level of NRF2 target genes were determined by real-time PCR. Treatment of compound **13b** significantly increased the expression of HO1, G6PD, and GCLC (Figure 4). This suggested that compound **13b** functionally activates NRF2 signaling. Moreover, we found that treatment of compound **13b** can effectively enhance the expression of NRF2 protein without altering the mRNA of NRF2 (Figure 5). This finding indicated that the possible mechanism of compound **13b** is the activation of NRF2 via the accumulation of NRF2 protein, rather than the promotion of the NRF2 transcription.

### 3.3. Molecular Docking

The molecular docking study of compound **13b**, the most potent NRF2 promoter, was carried out by the Achilles Blind Docking Server. Compound **13b** was docked with the Keap1-Kelch domain (PDB code 5FNQ) and the docking pose with the lowest binding energy was shown in Figure 6A. A channel-like shape in the middle of the structure of the Keap1-Kelch domain allowed compound **13b** to fit into the structure to form a stable complex. Once the channel was occupied by compound **13b**, the construction of the Keap1-Kelch domain would be distorted and would no longer interact with NRF2 resulting in the accumulation of the NRF2 protein. According to the docking results in Figure 6B, compound **13b** formed hydrophobic interactions with Val418, Val420, and Val561 and hydrogen bond with Gly367, Val 512, and Val616. The lowest binding energy score between compound **13b** and keap1 could reach to -9.60 kcal/mol. We speculated that the binding energy could reach the lowest score if the Michael addition interaction between the chalcone of compound **13b** and the cysteine of keap1 could be calculated.

## 4. Conclusions

We have synthesized certain 4-anilinoquinolinylchalcone derivatives and evaluated for their NRF2 activity. Among them, (*E*)-3-{4-[(4-acetylphenyl)- amino]quinolin-2-yl}-1-(4-methoxyphenyl) prop-2-en-1-one (**13b**) was found to significantly induce the NRF2 activity in HaCaT cells with a higher potency than that of *t*-BHQ and possessed no significant cell cytotoxicity (CC_50_ > 100 μM). The possible mechanism of compound **13b** is the activation of NRF2 via the accumulation of the NRF2 protein, rather than the promotion of the NRF2 transcription. Our results of this work might be helpful for discovering a novel class of potent NRF2 activators.

## 5. Experimental

### 5.1. General

All the chemical solvents and reagents used in this study were analytically pure without further purification and commercially available. Melting points were determined on an Electrothermal IA9100 melting point apparatus and are uncorrected. Nuclear magnetic resonance (^1^H and ^13^C) spectra were recorded on a Varian Gemini 200 (Palo Alto, CA, USA) spectrometer or Varian-Unity-400 spectrometer. Chemical shifts were expressed in parts per million (δ) with tetramethylsilane (TMS) as an internal standard. Thin-layer chromatography was performed on silica gel 60 F-254 plates purchased from E. Merck (Darmstadt, Germany). Compound purity is determined by high performance liquid chromatography (HPLC), and all final test compounds were > 95% purity. HPLC methods used the following: HITACHI Chromaster 5110 (Hitachi High-Technologies, Tokyo, Japan) with UV detection at 254 nm and auto sampler; column, Mightysil RP-18GP (250 mm × 4.6 mm, 5 μm, Kento chemical, Tokyo, Japan); mobile phase, method A: MeOH/0.01 M KH_2_PO_4_ (50:50), pH = 2.64; method B. MeOH/0.01M KH_2_PO_4_ (50:50), pH = 2.94; method C. MeOH/0.01M KH_2_PO_4_ (50:50), pH = 3.47; method D. MeOH/0.01M KH_2_PO_4_ (60:40), pH = 2.88; method E. MeOH/0.01M KH_2_PO_4_ (60:40), pH = 2.94; method F. MeOH/0.01M KH_2_PO_4_ (60:40), pH = 3.47; and method G. MeOH/0.01M KH_2_PO_4_ (80:20), pH = 3.47; and the flow rate was 1 mL/min; sample injection, 100 µM (1.0 mg dissolved in 0.5 mL DMSO, diluted with MeOH to 1 mL, then 0.5 mL of the compound was diluted with MeOH to 1 mL). The elemental analyses were performed in the Instrument Center of the National Science Council at National Cheng-Kung University and National Taiwan University using Heraeus CHN-O Rapid EA(Waltham, MA, USA), and all values were within ± 0.4% of the theoretical compositions.

### 5.2. General Procedure for the Preparation of 4-Chloroquinoline-2-Carbaldehyde **8**

A mixture of 4-hydroxy-2-methylquinoline **6** (3.18 g, 20 mmol) and POCl_3_ (10 mL) was heated at 140 °C for 2 h (TLC monitoring). After cooling, the reaction mixture was poured into ice-H_2_O (200 mL) and the concentrated NaOH solution was added until a pH of 10 was reached, and extracted with DCM (50 mL × 3). The organic layer was washed with brine, dried over MgSO_4_, and evaporated in vacuo to give crude product **7**, which was used without further purification in the next step. A mixture **7** (2.62 g, 15 mmol) and selenium dioxide (1.61 g, 15 mmol) in 1,4-dioxane (100 mL) was heated at 110 °C for 2 h (TLC monitoring). After cooling, the mixture was treated with 5% NaHCO_3_ aqueous (160 mL), extracted with DCM (100 mL × 3), the organic layer was collected, dried over MgSO_4_, and evaporated. The crude product was purified by flash chromatography on silica gel (DCM) and recrystallized by using EtOH to give **8**. Orange solid; yield: 70%; Mp. 139.8-140.2 °C ([22], lit: 138.1-139.2 °C); ^1^H-NMR (CDCl_3_) δ 10.18 (s, 1H, CHO), 8.32-8.26 (m, 2H), 8.09 (s, 1H, 3-H), 7.91-7.87 (m, 1H), 7.82-7.77 (m, 1H); ^13^C-NMR (CDCl_3_) δ 192.56, 152.34, 148.67, 144.20, 131.30, 130.85, 130.19, 128.05, 124.41, 117.57.

### 5.3. General Procedure for the Aldol Condensation between Acetophenone and **8**

To a suspension of appropriate 4-substituted acetophenone (10 mmol) with 3 M of NaOH (10 mL) in EtOH (30 mL) at 0 °C, was added to **8** (10 mmol, 1.91 g) carefully, and the mixture was stirred at 0 °C for 2 h (TLC monitoring). The mixture was neutralized with 6 N HCl until pH 7, the resulted precipitate thus obtained was collected, and the crude solid was recrystallized by using EtOH to give compounds **9–11.**

*(E)-3-(4-chloroquinolin-2-yl)-1-phenylprop-2-en-1-one (**9**)*, white solid; yield: 70%; Mp. 137.6-138.3 °C; ^1^H-NMR (CDCl_3_) δ 8.24-8.21 (m, 1H), 8.16-8.10 (m, 4H), 7.87 (d, 1H, J = 15.2 Hz, CH=CHC=O), 7.81 (m, 1H), 7.75 (s, 1H, 3-H), 7.68-7.60 (m, 2H), 7.55-7.51 (m, 2H); ^13^C-NMR (CDCl_3_) δ 190.25, 153.41, 149.06, 143.21, 142.37, 137.64, 133.21, 130.96, 130.23, 128.78 (2C), 128.73 (2C), 128.29, 127.77, 126.30, 124.07, 121.37; Anal. Calcd for C_18_H_12_ClNO: C 73.60, H 4.12, N 4.77; found: C 73.44, H 4.13, N 4.75.

*(E)-3-(4-chloroquinolin-2-yl)-1-(4-methoxyphenyl)prop-2-en-1-one (**10**)*, white solid; yield: 78%; Mp. 166.9-167.3 °C; ^1^H-NMR (CDCl_3_) δ 8.24-8.11 (m, 5H), 7.86 (d, 1H, *J* = 15.2 Hz, CH=CHC=O), 7.81 (m, 1H), 7.74 (s, 1H, 3-H), 7.66 (m, 1H), 7.03-6.99 (m, 2H), 3.90 (s, 3H, OCH_3_); ^13^C-NMR (CDCl_3_) δ 188.36, 163.79, 153.63, 149.07, 143.16, 141.51, 131.19 (2C), 130.91, 130.65, 130.19, 128.18, 127.72, 126.26, 124.07, 121.44, 113.97 (2C), 55.52; Anal. Calcd for C_19_H_14_ClNO_2_: C 70.48, H 4.36, N 4.33; found: C 70.15, H 4.48, N 4.37.

*(E)-3-(4-chloroquinolin-2-yl)-1-(4-fluorophenyl)prop-2-en-1-one (**11**)*, white solid; yield: 80%; Mp. 173.1-173.9 °C; ^1^H-NMR (CDCl_3_) δ 8.41 (d, 1H, *J* = 16.0 Hz, CH=CHC=O), 8.35 (s, 1H, 3-H), 8.31-8.28 (m, 2H), 7.96 (d, 1H, *J* = 16.0 Hz, CH=CHC=O), 7.86 (m, 1H), 7.80-7.74 (m, 2H), 7.17-7.11 (m, 2H); ^13^C-NMR (CDCl_3_) δ 188.34, 164.22 (*J*_CF_ = 250.1 Hz), 153.60, 147.98, 143.91, 143.83, 131.40 (*J*_CF_ = 3.0 Hz), 130.93, 130.88, 130.83 (2C, *J*_CF_ = 8.4 Hz), 129.59, 127.62, 124.25, 120.21, 120.19, 119.28, 115.65 (2C, *J*_CF_ = 21.2 Hz); Anal. Calcd for C_18_H_11_ClFNO: C 69.35, H 3.56, N 4.49; found: C 69.00, H 3.65, N 4.54.

### 5.4. General Procedure for the Preparation for 4-Substituted-Quinolinyl Chalcones **12a–12g**, **13a–13g**, and **14a–14g**

Compound **9**, **10**, or **11** (1.0 mmol) and appropriate *para*-substituted anilines (1.0 mmol) in EtOH/HCl (30/1 mL) was refluxed for 2 h (TLC monitoring). The mixture was then cooled and evaporated in vacuo to yield a residue, treated with ethyl acetate (50 mL), and the resulting precipitate was filtered and washed with ethyl acetate. The crude product was purified by flash chromatography on the silica gel (DCM/MeOH = 10/1), and recrystallized from EtOH to give 4-substituted-quinolinyl chalcones **12a–12g****.**

*(E)-1-phenyl-3-[4-(phenylamino)quinolin-2-yl]prop-2-en-1-one hydrochloride (**12a**)*, yellow solid; yield: 46%; Mp. 238.5-239.2 °C; HPLC (method D), t_R_ = 13.467 min; purity: 99.2%; ^1^H-NMR (DMSO-*d*_6_) δ 11.10 (br s, 1H, NH), 8.89-8.80 (m, 2H), 8.58 (m, 1H), 8.27 (m, 2H), 8.05 (m, 1H), 7.81-7.70 (m, 3H), 7.63-7.55 (m, 6H), 7.47-7.43 (m, 1H), 7.34 (s. 1H, 3-H); ^13^C-NMR (DMSO-*d*_6_) δ 188.53, 154.84, 147.47, 139.18, 137.03, 136.50, 134.65, 134.24, 134.06, 131.86, 129.91 (2C), 129.06 (2C), 128.99 (2C), 127.39, 127.12, 125.19 (2C), 123.45, 120.59, 116.91, 102.25; Anal. Calcd for C_24_H_18_N_2_O·1.2HCl: C 73.13, H 4.91, N 7.11; found: C 73.03, H 4.97, N 7.02.

*(E)-3-{4-[(4-acetylphenyl)amino]quinolin-2-yl}-1-phenylprop-2-en-1-one hydrochloride (**12b**)*, yellow solid; yield: 47%; Mp. 248.7-249.2 °C; HPLC (method F), t_R_ = 10.467 min; purity: 98.9%; ^1^H-NMR (DMSO-*d*_6_) δ11.17 (br s, 1H, NH), 8.90-8.83 (m, 2H), 8.58 (m, 1H), 8.27 (m, 2H), 8.13 (m, 2H), 8.07 (m, 1H), 7.84-7.59 (m, 8H), 2.64 (s, 3H, CH_3_); ^13^C-NMR (DMSO-*d*_6_) δ 196.87, 188.60, 148.40, 148.03, 142.90, 141.91, 136.51, 134.69, 134.60, 134.33, 134.09, 132.09, 129.99 (2C), 129.07 (2C), 129.02 (2C), 127.38, 124.00 (2C), 123.62, 120.91, 117.61, 103.14, 26.72; Anal. calcd for C_26_H_20_N_2_O_2_·1.1HCl: C 72.19, H 4.92, N 6.48; found: C 72.37, H 4.63, N 6.45.

*(E)-4-{[2-(3-oxo-3-phenylprop-1-en-1-yl)quinolin-4-yl]amino}benzoic acid hydrochloride (**12c**)*, yellow solid; yield: 51%; Mp.298.3-298.9 °C; HPLC (method F), t_R_ = 8.467 min; purity: 98.1%; ^1^H-NMR (DMSO-*d*_6_) δ 11.16 (br s, 1H, NH), 8.85-8.80 (m, 2H), 8.53 (m, 1H), 8.27-8.25 (m, 2H), 8.13-8.06 (m, 3H), 7.84-7.60 (m, 8H); ^13^C-NMR (DMSO-*d*_6_) δ 188.65, 166.76, 154.31, 147.94, 141.48, 139.23, 136.52, 134.55, 134.45, 134.15, 132.20, 131.08 (2C), 129.11 (2C), 129.07 (2C), 128.80, 127.46, 124.32 (2C), 123.63, 120.70, 117.49, 102.85; Anal. calcd for C_25_H_18_N_2_O_3_·1.1HCl: C 69.10, H 4.43, N 6.45; found: C 68.81, H 4.18, N 6.48.

*(E)-3-{4-{{4-{(E)-1-[(2-aminoethoxy)imino]ethyl}phenyl}amino}quinolin-2-yl)}-1-phenylprop-2-en-1-one hydrochloride (**12d**)*, yellow solid; yield: 38%; Mp. 200.2-201.5 °C; HPLC (method B), t_R_ = 8.613 min; purity: 98.4%; ^1^H-NMR (DMSO-*d*_6_) δ 11.29 (br s, 1H, NH), 9.00-8.90 (m, 2H), 8.67 (m, 1H), 8.31-8.29 (m, 4H), 8.06 (m, 1H), 7.89-7.59 (m, 9H), 7.46 (s, 1H, 3-H), 4.37 (t, 2H, *J* = 5.2 Hz), 3.18-3.16 (m, 2H), 2.32 (s, 3H, CH_3_); ^13^C-NMR (DMSO-*d*_6_) δ 188.61, 155.12, 154.47, 147.63, 139.37, 138.30, 136.53, 134.72, 134.24, 134.10 (2C), 132.04, 129.13 (2C), 129.02 (2C), 127.43 (2C), 127.16, 124.73 (2C), 123.76, 120.66, 117.22, 102.89, 69.91, 38.23, 12.71; Anal. calcd for C_28_H_26_N_4_O_2_·2.6HCl: C 61.67, H 5.29, N 10.27; found: C 61.32, H 4.95, N 10.27.

*(E)-3-{4-{{4-{(E)-1-{[3-(dimethylamino)propoxy]imino}ethyl}phenyl}-amino}quinolin-2-yl}-1-phenylprop-2-en-1-one hydrochloride (**12e**)*, yellow solid; yield: 52%; Mp. 155.6-156.6 °C; HPLC (method F), t_R_ = 7.647 min; purity: 96.1%; ^1^H-NMR (DMSO-*d*_6_) δ 9.26 (br s, 1H, NH), 8.41 (m, 1H), 8.13-8.07 (m, 3H), 8.00-7.98 (m, 1H), 7.79-7.67 (m, 5H), 7.62-7.56 (m, 4H), 7.47 (m, 2H), 4.17 (t, 2H, *J* = 6.4 Hz), 2.52-2.49 (m, 2H), 2.29 (s, 6H), 2.21 (s, 3H, CH_3_), 1.91-1.84 (m, 2H); ^13^C-NMR (DMSO-*d*_6_) δ 189.94, 153.52, 153.21, 148.92, 147.74, 144.47, 144.68, 137.38, 133.30, 130.60, 130.12, 129.65, 128.93 (C), 128.52 (2C), 126.97 (2C), 126.08, 125.63, 122.25, 120.83 (2C), 120.03, 103.68, 71.41, 55.41, 44.49 (2C), 26.34, 12.18; Anal. calcd for C_31_H_32_N_4_O_2_·0.3HCl: C 73.94, H 6.47, N 11.12; found: C 74.04, H 6.57, N 10.72.

*(E)-N-[2-(diethylamino)ethyl]-4-{[2-(3-oxo-3-phenylprop-1-en-1-yl)quino-lin-4-yl]amino}benzamide hydrochloride (**12f**)*, yellow solid; yield: 51%; Mp. 190.9-191.6 °C; HPLC (method E), t_R_ = 4.667 min; purity: 97.9%; ^1^H-NMR (DMSO-*d*_6_) δ 11.23 (br s, 1H, NH), 10.53 (br s, 1H, NH), 9.16 (m, 1H), 8.93-8.86 (m, 2H), 8.62 (m, 1H), 8.28 (m, 2H), 8.14 (m, 2H), 8.07 (m, 1H), 7.83-7.59 (m, 6H), 7.53 (s, 1H, 3-H), 3.71 (q, 2H, *J* = 6.0 Hz), 3.29-3.18 (m, 6H), 1.27 (t, 6H, *J* = 7.2 Hz); ^13^C-NMR (DMSO-*d*_6_) δ 188.58, 165.76, 154.27, 147.82, 140.27, 139.38, 136.51, 134.72, 134.28, 134.08, 132.03, 131.76, 129.09 (2C), 129.02 (4C), 127.27, 124.25 (2C), 123.68, 120.75, 117.38, 102.86, 49.95, 46.66 (2C), 34.26, 8.39 (2C); Anal. calcd for C_31_H_32_N_4_O_2_·4.5HCl: C 56.70, H 5.60, N 8.53; found: C 56.69, H 5.61, N 8.54.

*2-(Diethylamino)ethyl (E)-4-{[2-(3-oxo-3-phenylprop-1-en-1-yl)quinolin-4-yl]amino}benzoate hydrochloride (**12g**)*, yellow solid; yield: 52%; Mp. 158.7-159.3 °C; HPLC (method A), t_R_ = 6.640 min; purity: 98.9%; ^1^H-NMR (DMSO-*d*_6_) δ 11.31 (br s, 1H, NH), 10.71 (br s, 1H, NH), 8.93-8.89 (m, 2H), 8.62 (m, 1H), 8.29-8.19 (m, 4H), 8.07 (m, 1H), 7.84-7.71 (m, 5H), 7.63-7.59 (m, 3H), 4.68 (m, 2H), 3.56 (m, 2H), 3.25 (m, 4H), 1.30 (t, 6H, *J* = 7.2 Hz); ^13^C-NMR (DMSO-*d*_6_) δ 188.64, 164.86, 154.05, 148.04, 142.40, 139.50, 136.53, 134.67, 134.37, 134.16, 132.18, 131.23 (2C), 129.13 (2C), 129.06 (2C), 127.44, 126.76, 124.13 (2C), 123.86, 120.87, 117.71, 103.25, 59.48, 49.36, 47.03 (2C), 8.53 (2C); Anal. calcd for C_31_H_31_N_3_O_3_·3.6HCl: C 59.58, H 5.58, N 6.72; found: C 59.43, H 5.60, N 6.65.

*(E)-1-(4-Methoxyphenyl)-3-[4-(phenylamino)quinolin-2-yl]prop-2-en-1-one hydrochloride (**13a**)*, yellow solid; yield 48%; Mp. 220.6-221.1 °C; HPLC (method G), t_R_ = 4.467 min; purity: 99.1%; ^1^H-NMR (DMSO-*d*_6_) δ 11.05 (br s, 1H, NH), 8.85 (d, 1H, *J* = 15.6 Hz, CH=CHC=O), 8.78 (m, 1H), 8.56 (m, 1H), 8.27-8.24 (m, 2H), 8.04 (m, 1H), 7.78 (m, 1H), 7.68 (d, 1H, *J* = 15.6 Hz, CH=CHC=O), 7.62-7.54 (m, 4H), 7.44 (m, 1H), 7.31 (s, 1H, 3-H), 7.12-7.08 (m, 2H), 3.88 (s, 3H, OCH_3_); ^13^C-NMR (DMSO-*d*_6_) δ 186.53, 163.93, 154.80, 147.66, 139.17, 137.05, 134.20, 133.91, 132.06, 131.60 (2C), 129.93 (2C), 129.53, 127.39, 127.08, 125.18 (2C) 123.41, 120.59, 116.89, 114.29 (2C), 102.12, 55.70; Anal. calcd for C_25_H_20_N_2_O_2_·1.1HCl: C 71.40, H 5.06, N 6.66; found: C 71.42, H 4.94, N 6.42.

*(E)-3-{4-[(4-Acetylphenyl)amino]quinolin-2-yl}-1-(4-methoxyphenyl)prop-2-en-1-one hydrochloride (**13b**)*, yellow solid; yield: 50%; Mp. 248.7-249.2 °C; HPLC (method G), t_R_ = 6.087 min; purity: 99.0%; ^1^H-NMR (DMSO-*d*_6_) δ 11.09 (br s, 1H, NH), 8.85-8.78 (m, 2H), 8.53 (m, 1H), 8.26 (m, 2H), 8.15-8.06 (m, 3H), 7.84-7.71 (m, 4H), 7.62 (s, 1H, 3-H), 7.12 (m, 2H), 3.88 (s, 3H, OCH_3_), 2.64 (s, 3H, CH_3_); ^13^C-NMR (DMSO-*d*_6_) δ 196.94, 186.61, 163.98, 153.94, 148.29, 141.96, 139.49, 134.57, 134.31, 134.04, 132.25, 131.64 (2C), 130.03 (2C), 129.54, 127.39, 123.95 (2C), 123.57, 120.94, 117.63, 114.33 (2C), 103.03, 55.74, 26.75; Anal. Calcd for C_27_H_22_N_2_O_3_·1.5HCl: C 67.96, H 4.96, N 5.87; found: C 67.67, H 5.08, N 5.79.

*(E)-4-{{2-[3-(4-Methoxyphenyl)-3-oxoprop-1-en-1-yl]quinolin-4-yl}-amino}benzoic acid hydrochloride (**13c**)*, orange solid; yield: 49%; Mp. 298.3-298.9 °C; HPLC (method D), t_R_ = 9.900 min; purity: 96.7%; ^1^H-NMR (DMSO-*d*_6_) δ 11.14 (br s, 1H, NH), 8.87 (d, 1H, *J* = 16.0 Hz, CH=CHC=O), 8.80 (m, 1H), 8.57 (m, 1H), 8.27 (m, 2H), 8.12-8.04 (m, 3H), 7.82-7.68 (m, 4H), 7.59 (s, 1H, 3-H), 7.11 (m, 2H), 3.87 (s, 3H, OCH_3_); ^13^C-NMR (DMSO-*d*_6_) δ 186.53, 166.71, 163.94, 154.20, 148.05, 141.45, 139.20, 134.31, 133.75, 132.37, 131.62 (2C), 131.00 (2C), 129.50, 128.71, 127.32, 124.23 (2C), 123.57, 120.64, 117.41, 114.28 (2C), 102.77, 55.70; Anal. Calcd for C_26_H_20_N_2_O_4_·1.6HCl: C 64.68, H 4.51, N 5.80; found: C 64.61, H 4.62, N 5.53.

*(E)-3-{4-{{4-{(E)-1-[(2-Aminoethoxy)imino]ethyl}phenyl}amino}quinolin-2-yl}-1-(4-methoxyphenyl)prop-2-en-1-one hydrochloride (**13d**)*, orange solid; yield: 39%; Mp. 177.9-181.7 °C; HPLC (method B), t_R_ = 9.713 min; purity: 98.6%; ^1^H-NMR (DMSO-*d*_6_) δ 11.21 (br s, 1H, NH), 8.97 (d, 1H, *J* = 16.0 Hz, CH=CHC=O), 8.87 (m, 1H), 8.66 (m, 1H), 8.32-8.23 (m, 4H), 8.05 (m, 1H), 7.89-7.87 (m, 2H), 7.79 (m, 1H), 7.71 (d, 1H, *J* = 16.0 Hz, CH=CHC=O), 7.65-7.62 (m, 2H), 7.44 (s, 1H, 3-H), 7.13-7.10 (m, 2H), 4.37 (t, 2H, *J* = 5.6 Hz), 3.89 (s, 3H, OCH_3_), 3.18 (m, 2H), 2.32 (s, 3H, CH_3_); ^13^C-NMR (DMSO-*d*_6_) δ 186.60, 163.95, 155.12, 154.42, 147.82, 139.32, 138.30, 134.21, 134.06, 132.23, 131.67 (2C), 129.56, 127.44 (2C), 127.12, 124.70 (2C), 123.65, 120.66, 117.18, 114.30 (2C), 112.50, 102.79, 69.90, 55.73, 38.23, 12.70; Anal. calcd for C_29_H_28_N_4_O_3_·3.7HCl: C 56.59, H 5.19, N 9.10; Found: C 56.10, H 5.17, N 8.95.

*(E)-3-{4-{{4-{(E)-1-{[3-(Dimethylamino)propoxy]imino}ethyl}phenyl}-amino}quinolin-2-yl-1-(4-methoxyphenyl)prop-2-en-1-one hydrochloride (**13e**)*, orange solid; yield: 48%; Mp. 165.4-166.9 °C; HPLC (method C), t_R_ = 15.673 min; purity: 99.1%; ^1^H-NMR (DMSO-*d*_6_) δ 9.41 (br s, 1H, NH), 8.48 (m, 1H), 8.16-8.09 (m, 3H), 8.00 (m, 1H), 7.79-7.73 (m, 3H), 7.78 (d, 1H, *J* = 15.6 Hz, CH=CHC=O), 7.60-7.48 (m, 4H), 7.11 (m, 2H), 4.21 (t, 2H, *J* = 6.0 Hz), 3.88 (s, 3H, OCH_3_), 3.11 (m, 2H), 2.72 (s, 6H), 2.24 (s, 3H, CH_3_), 2.15-2.08 (m, 2H); ^13^C-NMR (DMSO-*d*_6_) δ 187.90, 163.39, 154.15, 153.27, 148.75, 147.85, 143.46, 141.87, 130.99 (2C), 130.38, 130.21, 129.41, 127.06 (2C), 126.12, 125.56, 122.50, 120.80 (2C), 120.03, 114.21 (2C), 103.71, 70.56, 55.63, 54.03, 42.09 (2C), 24.16, 12.32; Anal. Calcd for C_32_H_34_N_4_O_3_·4.8HCl: C 55.09, H 5.60, N 8.03; Found: C 54.75, H 5.56, N 7.91.

*(E)-N-[2-(Diethylamino)ethyl]-4-{{2-[3-(4-methoxyphenyl)-3-oxoprop-1-en-1-yl]quinolin-4-yl}amino}benzamide hydrochloride (**13f**)*, yellow solid; yield: 50%; Mp. 190.9-191.6 °C; HPLC (method B), t_R_ = 5.473 min; purity: 95.4%; ^1^H-NMR (DMSO-*d*_6_) δ 11.20 (br s, 1H, NH), 10.52 (br s, 1H, NH), 9.14 (m, 1H), 8.92 (d, 1H, *J* = 16.0 Hz, CH=CHC=O), 8.86 (m, 1H), 8.63 (m, 1H), 8.29 (m, 2H), 8.14 (m, 2H), 8.06 (m, 1H), 7.80 (m, 1H), 7.75-7.68 (m, 3H), 7.51 (s, 1H, 3-H), 7.12 (m, 2H), 3.88 (s, 3H, OCH_3_), 3.71 (m, 2H), 3.29-3.18 (m, 6H), 1.27 (t, 6H, *J* = 7.2 Hz); ^13^C-NMR (DMSO-*d*_6_) δ 186.54, 165.76, 163.94, 154.25, 147.97, 140.26, 139.31, 134.24, 133.92, 132.26, 131.75, 131.64 (2C), 129.53, 129.01 (2C), 127.22, 124.23 (2C), 123.64, 120.73, 117.33, 114.30 (2C), 102.76, 55.72, 49.95, 46.66 (2C), 34.27, 8.40 (2C)**;** Anal. calcd for C_32_H_34_N_4_O_3_·3.0HCl: C 60.81, H 5.90, N 8.86; found: C 60.81, H 6.05, N 8.86.

*2-(Diethylamino)ethyl (E)-4-{{2-[3-(4-methoxyphenyl)-3-oxoprop-1-en-1-yl]-quinolin-4-yl}amino}benzoate hydrochloride (**13g**)*, yellow solid; yield: 53%; Mp. 158.7-159.3 °C; HPLC (method A), t_R_ = 7.553 min; purity: 99.3%; ^1^H-NMR (DMSO-*d*_6_) δ 10.75 (br s, 1H, NH), 8.92-8.84 (m, 2H), 8.61 (m, 1H), 8.29 (m, 2H), 8.19 (m, 2H), 8.05 (m, 1H), 7.82-7.73 (m, 4H), 7.62 (s, 1H, 3-H), 7.12 (m, 2H), 4.68 (m, 2H), 3.89 (s, 3H, OCH_3_), 3.55 (m, 2H), 3.26-3.24 (m, 4H), 1.29 (t, 6H, *J* = 7.2 Hz); ^13^C-NMR (DMSO-*d*_6_) δ 186.65, 164.83, 163.92, 153.48, 148.55, 142.70, 134.61, 133.99, 131.95, 131.62 (2C), 131.19 (2C), 129.59, 127.23, 126.33, 123.69 (2C), 121.48, 117.88, 116.03, 115.81, 114.30 (2C), 103.46, 59.39, 55.72, 49.30, 46.95 (2C), 8.48 (2C); Anal. calcd for C_32_H_33_N_3_O_4_·3.3HCl: C 59.68, H 5.68, N 6.52; found: C 59.34, H 5.81, N 6.33.

*(E)-1-(4-Fluorophenyl)-3-[4-(phenylamino)quinolin-2-yl]prop-2-en-1-one hydrochloride (**14a**)*, yellow solid; yield: 54%; Mp. 231.7-232.1 °C; HPLC (method E), t_R_ = 11.213 min; purity: 98.8%; ^1^H-NMR (DMSO-*d*_6_) δ 11.09 (br s, 1H, NH), 8.87 (d, 1H, *J* = 15.6 Hz, CH=CHCO), 8.79 (m, 1H), 8.57 (m, 1H), 8.39-8.34 (m, 2H), 8.05 (m, 1H), 7.78 (m, 1H), 7.74 (d, 1H, *J* = 15.6 Hz, CH=CHCO), 7.61-7.54 (m, 4H), 7.46-7.40 (m, 3H), 7.33 (s, 1H, 3-H); ^13^C-NMR (DMSO-*d*_6_) δ 187.08, 165.55 (*J*_CF_ = 251.6 Hz), 154.90, 147.39, 139.12, 137.01, 134.72, 134.31, 133.28 (*J*_CF_ = 2.7 Hz), 132.20 (2C, *J*_CF_ = 9.9 Hz), 131.72, 129.94 (2C), 127.45, 127.17, 125.22 (2C), 123.46, 120.52, 116.90, 116.17 (2C, *J*_CF_ = 21.8 Hz), 102.29; Anal. calcd for C_24_H_17_FN_2_O·1.6HCl: C 67.55, H 4.39, N 6.56; found: C 67.40, H 4.43, N 6.68.

*(E)-3-{4-[(4-Acetylphenyl)amino]quinolin-2-yl}-1-(4-fluorophenyl)prop-2-en-1-one hydrochloride (**14b**)*, yellow solid; yield: 52%; Mp. 268.8-269.5 °C; HPLC (method G), t_R_ = 6.087 min; purity: 99.1%; ^1^H-NMR (TFA-*d*) δ 8.36 (m, 1H), 8.24 (m, 2H), 8.06-7.97 (m, 5H), 7.83-7.76 (m, 2H), 7.62 (m, 2H), 7.42 (s, 1H, 3-H), 7.17 (m, 2H), 2.75 (s, 3H, CH_3_); ^13^C-NMR (TFA-*d*) δ 202.37, 189.83, 165.99 (*J*_CF_ = 259.3 Hz), 153.70, 145.54, 139.72, 136.71, 133.83, 133.49, 133.21, 130.55 (2C, *J*_CF_ = 9.9 Hz), 129.97, 129.66, 129.30 (2C), 127.10, 122.80 (2C), 119.63, 118.37, 115.39, 114.49 (2C, *J*_CF_ = 22.8 Hz), 98.74, 22.75; Anal. calcd for C_26_H_19_FN_2_O_2_·1.5HCl: C 67.14, H, 4.44, N 6.02; found: C 66.77, H 4.67, N 5.98.

*(E)-4-({2-[3-(4-Fluorophenyl)-3-oxoprop-1-en-1-yl]quinolin-4-yl}amino)-benzoic acid hydrochloride (**14c**)*, yellow solid; yield: 49%; Mp. 278.9-280.2 °C; HPLC (method F), t_R_ = 10.780 min; purity: 99.3%; ^1^H-NMR (DMSO-*d*_6_) δ 13.02 (br s, 1H, OH), 11.07 (br s, 1H, NH), 8.83-8.77 (m, 2H), 8.51 (m, 1H), 8.36-8.33 (m, 2H), 8.12-8.04 (m, 3H), 7.83-7.78 (m, 2H), 7.68 (m, 2H), 7.62 (s, 1H, 3-H), 7.44 (m, 2H); ^13^C-NMR (DMSO-*d*_6_) δ 187.17, 166.82, 165.52 (*J*_CF_ = 242.8 Hz), 154.02, 148.04, 141.56, 139.44, 134.94, 134.31, 133.29 (*J*_CF_ = 2.6 Hz), 132.18 (2C, *J*_CF_ = 9.6 Hz), 131.79, 131.06 (2C), 128.64, 127.40, 124.11 (2C), 123.54, 120.97, 117.56, 116.14 (2C, *J*_CF_ = 21.8 Hz), 102.90; Anal. calcd for C_25_H_17_FN_2_O_3_·1.7HCl: C 63.29, H 3.97, N 5.90; found: C 63.02, H 4.12, N 5.83.

*(E)-3-{4-{{4-{(E)-1-[(2-Aminoethoxy)imino]ethyl}phenyl}amino}quinolin-2-yl}-1-(4-fluorophenyl)prop-2-en-1-one hydrochloride (**14d**)*, yellow solid; yield: 40%; Mp.180.1-180.8 °C; HPLC (method A), t_R_ = 8.660 min; purity: 99.2%; ^1^H-NMR (DMSO-*d*_6_) δ 11.25 (br s, 1H, NH), 9.00-8.88 (m, 2H), 8.66 (m, 1H), 8.41-8.28 (m, 5H), 8.04 (m, 1H), 7.87 (m, 2H), 7.78-7.74 (m, 2H), 7.63 (m, 2H), 7.45-7.41 (m, 3H), 4.37 (m, 2H), 3.17 (m, 2H), 2.32 (s, 3H, CH_3_); ^13^C-NMR (DMSO-*d*_6_) δ 187.10, 165.54 (*J*_CF_ = 251.7 Hz), 155.10, 154.34, 147.63, 139.48, 138.32, 134.94, 134.15, 134.01, 133.29 (*J*_CF_ = 2.3 Hz), 132.22 (2C, *J*_CF_ = 9.1 Hz), 131.70, 127.40 (2C), 127.11, 124.64 (2C), 123.69, 120.78, 117.21, 116.09 (2C, *J*_CF_ = 21.9 Hz), 102.95, 69.88, 38.21, 12.70; Anal. calcd for C_28_H_25_FN_4_O_2_·1.9HCl: C 62.53, H 5.04, N 10.42, found: C 62.71, H 5.02, N 10.53.

*(E)-3-{4-{{4-{(E)-1-{[2-(Dimethylamino)ethoxy]imino}ethyl}phenyl}-amino}quinolin-2-yl}-1-(4-fluorophenyl)prop-2-en-1-one hydrochloride (**14e**)*, yellow solid; yield: 51%; Mp.131.2-132.7 °C; HPLC (method E), t_R_ = 4.793 min; purity: 97.7%; ^1^H-NMR (DMSO-*d*_6_) δ 9.26 (br s, 1H, NH), 8.41 (m, 1H), 8.17 (m, 2H), 8.09 (d, 1H, J = 15.6 Hz, CH=CHCO), 7.98 (m, 1H), 7.78-7.69 (m, 4H), 7.60-7.56 (m, 2H), 7.47-7.38 (m, 4H), 4.17 (t, 2H, J = 6.0 Hz), 2.66 (m, 2H), 2.40 (s, 6H), 2.21 (s, 3H, CH_3_), 1.95-1.89 (m, 2H); ^13^C-NMR (DMSO-*d*_6_) δ 188.45, 165.09 (J_CF_ = 251.0 Hz), 153.68, 153.17, 148.91, 148.71, 144.60, 141.73, 134.04 (*J*_CF_ = 3.1 Hz), 131.56 (2C, *J*_CF_ = 9.8 Hz), 130.50, 130.12, 129.62, 126.99 (2C), 125.84, 125.62, 122.27, 120.76 (2C), 120.04, 115.94 (2C, *J*_CF_ = 22.0 Hz), 103.72, 71.16, 55.05, 43.86 (2C), 25.76, 12.20; Anal. calcd for C_31_H_31_FN_4_O_2_·1.5HCl: C 65.86, H 5.79, N 9.91. Found: C 65.48, H 5.93, N 9.86.

*(E)-N-[2-(Diethylamino)ethyl]-4-{{2-[3-(4-fluorophenyl)-3-oxoprop-1-en-1-yl]quinolin-4-yl}amino}benzamide hydrochloride (**14f**)*, yellow solid; yield: 50%; Mp. 175.6-176.1 °C; HPLC (method A), t_R_ = 5.847 min; purity: 99.1%; ^1^H-NMR (DMSO-*d*_6_) δ 11.31 (br s, 1H, NH), 10.66 (br s, 1H, NH), 9.19 (m, 1H), 9.01-8.91 (m, 2H), 8.68 (m, 1H), 8.40 (m, 2H), 8.16-8.05 (m, 3H), 7.82-7.68 (m, 4H), 7.53 (s, 1H, 3-H), 7.47-7.43 (m, 2H), 3.71 (m, 2H), 3.29-3.17 (m, 6H), 1.27 (t, 6H, *J* = 7.2 Hz); ^13^C-NMR (DMSO-*d*_6_) δ 187.12, 165.74, 165.55 (*J*_CF_ = 251.7 Hz), 154.29, 147.76, 140.30, 139.47, 134.85, 134.23, 133.31 (*J*_CF_ = 3.0 Hz), 132.24 (2C, *J*_CF_ = 9.0 Hz), 131.83, 131.72, 129.00 (2C), 127.23, 124.22 (2C), 123.80, 120.75, 117.41, 116.13 (2C, *J*_CF_ = 22.0 Hz), 103.05, 49.94, 46.65 (2C), 34.24, 8.40 (2C); Anal. Calcd for C_31_H_31_FN_4_O_2_·3.6HCl: C 58.01, H 5.43, N 8.73; found: C 57.84, H 5.44, N 8.93.

*2-(Diethylamino)ethyl (E)-4-{{2-[3-(4-fluorophenyl)-3-oxoprop-1-en-1-yl]-quinolin-4-yl}amino}benzoate hydrochloride (**14g**)*, yellow solid; yield: 50%; Mp. 135.4-135.8 °C; HPLC (method A), t_R_ = 8.073 min; purity: 98.8%; ^1^H-NMR (DMSO-*d*_6_) δ 9.60 (br s, 1H, NH), 8.41 (m, 1H), 8.23-8.14 (m, 3H), 8.04-8.01 (m, 3H), 7.81-7.76 (m, 3H), 7.62 (m, 1H), 7.54 (m, 2H), 7.45-7.40 (m, 2H), 4.53 (br s, 2H), 3.04 (br s, 4H), 1.20 (m, 6H); ^13^C-NMR (DMSO-*d*_6_) δ 188.35, 165.15, 165.14 (*J*_CF_ = 250.2 Hz), 153.37, 148.99, 146.60, 146.47, 144.35, 134.01 (*J*_CF_ = 3.0 Hz), 131.64 (2C, *J*_CF_ = 9.1 Hz), 131.12 (2C), 130.34, 129.65, 126.11, 122.60, 122.21, 120.84, 118.65 (2C), 115.97 (2C, *J*_CF_ = 22.0 Hz), 106.09, 60.19, 49.78, 47.00 (2C), 9.46 (2C); Anal. calcd for C_31_H_30_FN_3_O_3_·4.1HCl: C 56.32, H 5.20, N 6.36; found: C 56.21, H 5.49, N 6.29.

### 5.5. Cytotoxicity and Antiviral Activity Assays

#### 5.5.1. Compounds

Compounds were dissolved in DMSO at 10 mM and then diluted in the culture medium.

#### 5.5.2. Cell Culture

Immortalized human keratinocytes HaCaT were cultured in the Dulbecco’s modified Eagle’s medium (DMEM; Gibco BRL, Grand Island, NY, USA) with 10% heat-inactivated fetal bovine serum (HyClone, Logan, UT, USA), penicillin (100 U/mL), and streptomycin (100 μg/mL) in a humidified incubator with 5% CO_2_. HaCaT/ARE stable cells were grown in DMEM supplemented with 100 μg/mL of hygromycin.

#### 5.5.3. Luciferase Reporter Assay

HaCaT/ARE stable cells were seeded (1 × 10^4^ cells/well) in a 96 well plate, then were treated with indicated concentrations of compounds for 18 h. At the assay time point, resazurin (Cayman Chemical, Ann Arbor, MI, USA) was added to a final concentration of 0.1 mg/mL and further incubated for 4 h at 37 °C. Fluorescence of the reduced resazurin (ex/em: 530 nm/590 nm) was measured from the culture supernatant by using a Synergy HT Multi-Mode Reader (BioTek, Winooski, VT, USA) to determine cell viability. The cells were then harvested for luciferase activity measurements according to the manufacturer’s protocol (Promega Corporation, Madison, WI, USA). Relative luciferase activity was calculated by normalizing luciferase activity to cell viability, and DMSO solvent control was used as 100% activity.

#### 5.5.4. Cell Viability Assay

Cells were seeded in a 96-well plate contained 100 μL culture medium in triplicate and treated with several compounds for 72 h. At the assay time point, resazurin (Cayman Chemical) was added and further incubated for 4 h at 37 °C. Fluorescence of the reduced resazurin was measured as described previously.

#### 5.5.5. Quantitative Real-Time PCR (qRT-PCR)

RNA was prepared by using the TRIzol Reagent (Thermo Scientific, Madison, WI, USA) and was reverse transcribed into cDNA using a TOOLs Easy Fast RT Kit (TOOLs Biotechnology, New Taipei City, Taiwan). PCR was performed on an ABI StepOne Plus System (Applied Biosystems, Foster City, CA, USA) using the KAPA SYBR^®^ FAST qPCR Master Mix (2X) Kit (Sigma-Aldrich, St. Louis, MO, USA). The mRNA level was normalized using the glyceraldehyde 3-phosphate dehydrogenase (GAPDH) mRNA level as the standard. The following primers were used: NRF2 forward: 5′-AGACGGTATGCAACAGGACA and reverse: 5′-ACCATGGTAGTCTCAACCAGC; HO1 forward: 5′-GCCAGCAACAAAGTGCAAG and reverse: 5′-GAGTGTAAGGACCCATCGGA; NQO1 forward: 5′-TGCAGCGGCTTTGAAGAAGAAAGG and reverse: 5′-TCGGCAGGATACTGAAAGTTCGCA; GCLC forward: 5′-CTGGGGAGTGATTTCTGCAT and reverse: 5′-AGGAGGGGGCTTAAATCTCA; G6PD forward: 5′-CAACATCGCCTGCGTTA and reverse: 5′-CTTGACCTTCTCATCACGG; and GAPDH forward: 5′-GCAAATTCCATGGCACCGTCA and reverse: 5′-TCCTGGAAGATGGTGATGGGA.

#### 5.5.6. Western Blot

Cells were washed with PBS and harvested with RIPA lysis buffer (50 mM Tris (pH 7.5), 150 mM NaCl, 1% Tx-100) supplemented with protease and phosphatase inhibitors (1 mM PMSF, 10 μg/mL leupeptin, 50 μg/mL TLCK, 50 μg/mL TPCK, 1 μg/mL aprotinin, 1 mM NaF, 5 mM NaPPi, and 10 mM Na_3_VO_4_). The cell lysates were cleared via 10 min spins at 13,000 rpm, 4 °C. Protein concentrations were determined by a Bio-Rad Protein Assay (Bio-Rad, Hercules, CA, USA). Samples corresponded to 20 μg of protein were used for Western blot according to our previous publication [7]. The following antibodies were used: anti-NRF2 (GeneTex, Irvine, CA, USA) and anti-GAPDH (Abcam, Cambridge, MA, USA).

#### 5.5.7. Molecular Docking Study

The crystal structure of the Keap1 Kelch domain (PDB code 5FNQ) [27] was acquired from the RCSB Protein Date Bank. The 3D conformation of target compound **13b** was produced by ChemBio 3D Ultra 14.0. The molecular docking was performed by the Achilles Blind Docking Server (http://bio-hpc.ucam.edu/achilles/). The “blind docking” approach was used for the docking of the small molecule to the targets, which was done without a priori knowledge of the location of the binding site by the system [28]. Visual representation of molecules was created with 3Dmol by Nicholas Rego and David Koes [29].

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
