# Peer review of "Discovery of 4-Anilinoquinolinylchalcone Derivatives as Potential NRF2 Activators"

_molecules, 2020, doi:10.3390/molecules25143133_

Round 1
Reviewer 1 Report
The manuscript by Kao et al describes the synthesis and biological evaluation of a new series of 4-anilinoquinolinylchalcone derivatives as activators of NRF2. The new molecules were appropriately characterized and have adequate purity. The authors provided the corresponding charts to support the structural characterization and the purity of molecules. All other experimental details pertaining to the biological evaluation are also well described. The study has identified a reasonable and nontoxic lead i.e. 13b with appropriate activity for subsequent development as anticancer and/or chemo-preventive agent. The manuscript is also generally well organized. Figures and tables are appropriate. Minor English errors are to be fixed. Overall, I recommend publishing the manuscript in the present form.
Author Response
Thank you for your comments. We have carefully checked English of the revised manuscript.

Reviewer 2 Report
I think overall it is a great paper
Review comments on “Discovery of 4-anilinoquinolinylchalaone derivatives as potential NRF2 activators”
Kao and team have synthesized a series of 4-anilinoquinolinylchalaone derivatives and used NRF2
promoter driven firefly luciferase reporter stable HaCaT/ARE cells to screen for the appropriate
molecule. Among the derivatives, 13b shows the most promising effects. This does shows potential in
helping the discovery of potent NRF2 activators.
I do have a few comments on the paper:
My first concern is that fig 2 to fig 6 in my pdf versions are all out of proportion and I am unable to fully
see all of the results. I am unsure why it turn out to be like this. Here are some examples I am seeing.
And for the ARE-Luciferase activity test, shouldn’t all 3 moleclues have the same concentration instead?
As it is seen to be tested at different concentration.
Overall, I think this paper is simple and straight forward. I am just confused about why the figures are
out of proportion for my pdf and unable to see the whole thing.
Additional comments on “Discovery of 4-Anilinoquinolinylchalaone Derivatives as Potential NRF2 Activators”
In this paper, Kao and team understood the importance of NRF2 activator in prevention of cancer development, and they were able to synthesize a series of molecules and further tested their efficiency via luciferase with HaCaT/ARE cells, and among the derivatives, compound 13b stood out.
I think this paper is straight forward in the discovery of NRF2 activator molecule, and the steps for this discovery is reasonable. The novelty of this discovery I feel is fairly low because there are already several known molecules as NRF2 activators (Fig.1.), as from table.1 you can see that the positive control t-BHQ has drastically higher cell viability and similar NRF2 activity in HaCaT cell to 13b.
However, there are several points I would like to see in this paper that could potentially increase the quality.
1. Since in fig.1 that several of the known NRF2 activators are presented, if possible, I would like to see the comparison towards the target compounds.
2. Since t-BHQ acted as a positive control for % of relative NRF2 activity, I think it will be meaning full to put t-BHQ into rest of the test as a comparison towards 13b, like the cell viability, ARE-luciferase activity% and increase expression of NRF2 target genes.
3. For Fig. 2, it was briefly mentioned on the activity%, it seems that after 25uM, even 13b starts to decrease in activity, is there an explanation for that?
As I previously mentioned, this is good paper as it is straight forward towards its discovery, meaning they know that NRF2 activator is their target, and made several derivatives for the activation and using different methods to single out which molecule is the most efficient. However, I do not see the novelty in this paper because there are already several other molecules been discovered, unless they compared them with their own discovery.
Author Response
We have provide a point-by-point response to the reviewer’s comments and upload it as a Word/PDF file.
